# Accurate Measurement of Copper Overload in an Experimental Model of Wilson Disease by Laser Ablation Inductively Coupled Plasma Mass Spectrometry

**DOI:** 10.3390/biomedicines8090356

**Published:** 2020-09-16

**Authors:** Philipp Kim, Chengcheng Christine Zhang, Sven Thoröe-Boveleth, Sabine Weiskirchen, Nadine Therese Gaisa, Eva Miriam Buhl, Wolfgang Stremmel, Uta Merle, Ralf Weiskirchen

**Affiliations:** 1Institute of Molecular Pathobiochemistry, Experimental Gene Therapy and Clinical Chemistry (IFMPEGKC), RWTH University Hospital Aachen, D-52074 Aachen, Germany; hkim@ukaachen.de (P.K.); sweiskirchen@ukaachen.de (S.W.); 2Department of Internal Medicine IV, Heidelberg University Hospital, D-69120 Heidelberg, Germany; ChengchengChristine.Zhang@med.uni-heidelberg.de (C.C.Z.); Uta.Merle@med.uni-heidelberg.de (U.M.); 3Institute for Occupational, Social and Environmental Medicine, RWTH Aachen University, D-52074 Aachen, Germany; sven.thoroe@rwth-aachen.de; 4Institute of Pathology, RWTH Aachen University, D-52074 Aachen, Germany; ngaisa@ukaachen.de; 5Electron Microscopy Facility, Institute of Pathology, RWTH Aachen University Hospital, D-52074 Aachen, Germany; ebuhl@ukaachen.de; 6Medical Center Baden-Baden, D-76530 Baden-Baden, Germany; wolfgangstremmel@aol.com

**Keywords:** Wilson disease, copper overload, metals, mass spectrometry, rhodanine, diagnostic, LA-ICP-MS, quantification, electron microscopy, scanning electron microscopy

## Abstract

Wilson disease is a rare inherited autosomal recessive disorder. As a consequence of genetic alterations in the *ATP7B* gene, copper begins to accumulate in the body, particularly in the liver and brain. Affected persons are prone to develop liver cancer and severe psychiatric and neurological symptoms. Clinically, the development of corneal Kayser-Fleischer rings and low ceruloplasmin concentrations (<20 mg/dL) are indicative of Wilson disease. However, the detection of elevated hepatic copper content (>250 µg/g dry weight) alone is still considered as the best but not exclusive diagnostic test for Wilson disease. Presently, specific copper stains (e.g., rhodanine) or indirect staining for copper-associated proteins (e.g., orcein) are widely used to histochemically visualize hepatic copper deposits. However, these procedures only detect lysosomal copper, while cytosolic copper is not detectable. Similarly, elemental analysis in scanning electron microscope with energy dispersive X-ray analysis (EDX) often leads to false negative results and inconsistencies. Here, we tested the diagnostic potential of laser ablation inductively-coupled mass spectrometry (LA-ICP-MS) that allows quantitative analysis of multiple elements. Comparative studies were performed in wild type and the *Atp7b* null mouse model. We propose LA-ICP-MS as a versatile and powerful method for the accurate determination of hepatic copper in people with Wilson disease with high spatial resolution.

## 1. Introduction

Wilson disease (WD) is an inherited progressive disorder characterized by excess copper stored in various body tissues, particularly in the liver and the brain [1]. Untreated, it results in hepatic disease and central nervous system dysfunction. Clinical evaluation, patient’s family history, and specialized genetic and blood (serum) tests are helpful in the diagnosis of WD. However, in most patients, a liver biopsy for copper analysis is necessary to confirm the WD diagnosis. The demonstration of hepatic copper or copper-associated proteins is possible by a large number of histochemical stains. However, it is known that the histochemical evaluation of copper might result in negative results because the intracellular distribution of copper changes in relation to the stage of the disease. In early stages, copper is diffusely distributed in the hepatocyte cytoplasm and the metal is only hardly detectable in rhodanine stain [2]. At later stages, copper is located both within the cytoplasm as well as concentrated in the lysosomal compartment. Lysosomal copper is more readily identifiable and best demonstrable in rhodanine stain [3]. In the pre-cirrhotic stage, copper is found in periportal hepatocytes, while in fully developed cirrhosis, the pattern of copper is typically uneven, with only some nodules staining positive and some staining even more negative. At these stages, the rhodanine stain has a minor value in the diagnosis of early and late phases of Wilson disease [4]. Therefore, the pathological determination of copper is highly challenging, and a negative copper stain does not exclude the possibility of Wilson disease.

Laser ablation inductively-coupled mass spectrometry (LA-ICP-MS) is suitable for bioimaging the elemental composition in biological samples. In this technique, a finely focused laser beam ablates material from a biological sample that is vaporized, atomized, and ionized in an inductively-coupled plasma atmosphere. Subsequently, the ions are extracted into a mass spectrometer, separated according to their mass to charge ratio, and the obtained signal is quantified [5]. Most recently, we have used this technique to demonstrate the therapeutic efficacy of a gene correction approach in WD in brain and liver [6,7]. These studies have already demonstrated that LA-ICP-MS is a powerful and sensitive technique, enabling simultaneous imaging and quantification of a variety of trace metals which good sensitivity for major, minor, trace and ultra-trace elements [7]. However, in these former studies we have not compared our novel LA-ICP-MS protocols with other techniques commonly used in WD diagnostics. To do so, we have now used the *Atp7b*^–/–^ mouse model and evaluated features of hepatic copper overload by histochemistry, scanning electron microscopy with energy–disperse X-ray (EDX) spectroscopy analysis (SEM-EDX), electron microscopic analysis, and LA-ICP-MS. We show that LA-ICP-MS is the most versatile method for diagnosis of copper overload. Compared to all other methods, it provides quantitative data on absolute copper concentrations and further is suitable to simultaneously identify subtle changes in other trace elements. Therefore, we suggest LA-ICP-MS as a very powerful method that will revolutionize WD diagnostics in the near future.

## 2. Experimental Section

### 2.1. Animals

The characteristics of the *Atp7b* null mutation mice model used in this study were described elsewhere [8]. In brief, the *Atp7b* gene was disrupted by insertion in exon 2 of multiple stop codons covering all possible reading frames. The resulting transgenic mice express a modified *Atp7b* mRNA, which is translated into considerable smaller non-functional proteins. Mutant mice and age-matched wild type controls had a genetic 129/Sv background and were housed at the University of Heidelberg, according to the guidelines of the Institutional Animal Care and Use Committees and in accordance with governmental requirements [9]. Ethical approval of animal experiments was given on August 4, 2015 under permit number 35-9185.81/G-74/15. In total, 16 male mice (11 wild type, 5 *Atp7b*^–/–^) were analyzed in this study. Mice were housed under controlled temperature conditions and relative humidity, with 10–15 air changes per hour and light illumination for 12 h per day and free access to food and water throughout the acclimation and experimental phase.

### 2.2. Rhodanine Stain for Detection of Cytoplasmic Accumulation of Hepatic Copper

Formalin-fixed tissue was stained with rhodanine following standard protocols [10]. Following staining, the slides were rinsed in dH_2_O and counterstained in hematoxylin for 1 min. As a positive control, a formalin-fixed histologic liver section of a cirrhotic Wilson disease patient was taken.

### 2.3. Scanning Electron Microscopy with Energy-Disperse X-ray (EDX) Spectroscopy Analysis (SEM-EDX)

Small liver cryo-slices of 20 to 50 µm thickness were sputtered with a thin conductive silver paint, carbon-coated in a rotary evaporator, and then maintained in desiccators to prevent air contact before analysis. EDX analysis of elements was performed within two days of preparation using an environmental XL30 ESEM-FEG scanning electron microscope (Philips, Eindhoven, The Netherlands) in a high vacuum environment operating at 15 to 20 kV. The slices were imaged by backscattered electrons and analyzed for the elemental composition of the element present. For quantitative analysis of the element concretions, the contributions of the carbon-coating and the embedding resin containing carbon (C), oxygen (O) and traces of chloride (Cl) were computer-based subtracted from the quantitative data of each spectrum.

### 2.4. Electron Microscopic Analysis

Directly after dissection of the livers, tissue was cut into small pieces (1–2 mm) and fixed in 3% glutaraldehyde in 1× phosphate buffered saline. After washing in 0.1 M Soerensen’s phosphate buffer (Merck, Darmstadt, Germany), the samples were post-fixed in 1% Osmium tetroxide (OsO_4_) (Roth, Karlsruhe, Germany) solved in 17% sucrose buffer (Merck) and dehydrated by ascending ethanol series (30%, 50%, 70%, 90%, and 100%) for 10 min each. The last step was repeated three times. Subsequently, dehydrated specimens were incubated in propylene oxide (Serva, Heidelberg, Germany) for 30 min, in a mixture of Epon resin (Serva) and propylene oxide (1:1) for 1 h, and finally, in pure Epon for 1 h. Epon polymerization was performed at 90 °C for 2 h. Finally, ultrathin sections (70–100 nm) were cut with an ultramicrotome (Reichert Ultracut S, Leica, Wetzlar, Germany) using a diamond knife (Diatome Ltd., Nidau, Switzerland) and picked up on Cu/Rh grids (HR23 Maxtaform, Plano GmbH, Wetzlar, Germany). Samples were viewed without additional contrast staining at an acceleration voltage of 60 kV using a Zeiss Leo 906 (Carl Zeiss AG, Oberkochen, Germany) transmission electron microscope. Pictures were acquired at magnifications of 46,460× and 60,000×, respectively.

### 2.5. Sample Preparation for LA-ICP-MS Measurements

Liver samples were cryo-cut into 30 μm thick slices with a CM3050S cryomicrotome (Leica Biosystems, Wetzlar, Germany) on –18 °C cryo-chamber temperature and –16 °C object area temperature, and thaw-mounted onto adhesive StarFrost^®^ microscope slides (Knittel Glass, Braunschweig, Germany). Samples were dried and stored at room temperature until analysis.

### 2.6. LA-ICP-MS Set Up and Measurements

Prior measurement, the mounted tissues were first scanned in a NanoZoomer-SQ digital slide scanner (Hamamatsu Photonics Germany GmbH, Herrsching am Ammersee, Germany). The LA-ICP-MS measurements were performed in a system combined of a high performing triple quadrupole Agilent 8900 ICP-MS (Agilent Technologies, Santa Clara, CA, USA) and a New Wave NWR213 laser ablating device (Elemental Scientific, Omaha, NE, USA). The equipment was installed in a conditioned room (Figure 1). The sample chamber of the laser ablation was subjected to a constant flow of helium gas, so that external contamination of the samples is considered unlikely.

Standards for the determination of element concentrations were produced from homogenized tissue spiked with varying concentrations of a standard salt solution. The parameters used in the measurements were: Radiofrequency (RF) power input: 1450 W, cooling gas flow rate: 16 L/min, auxiliary gas flow rate: 0–7 L/min, carrier gas flow rate: 1.0 L/min, Dwell time: 20 ms, extraction lens potential: 3400 V, mass resolution: 300 m/Δm, scanning mode: peak hopping, scanning speed: 50–60 µm/sec, ablation mode: line scan, repetition frequency: 20 Hz, laser fluence: 0.24 J/cm^2^. This resulted in a typical analysis time of 4–6 h per sample.

### 2.7. Image Generation of Bio-Metal Distribution

Isotope images were generated in Microsoft Excel using the in-house generated Excel Laser Ablation Imaging (ELAI) visualization tool described before [11,12]. This software allows simple image generation from mass spectrometry data and is freely available [11].

## 3. Results

### 3.1. Electron Microscopic Analysis of Atp7b^–/–^ Mouse Liver Tissues

It is well accepted that copper overload in liver cells leads to multiple cellular lesions [13]. In particular, significant ultrastructural changes occur during progression of WD. These include severe mitochondrial changes, increased numbers of peroxisomes and cytoplasmic lipid droplets, and characteristic cytoplasmic bodies formed by lipid vacuoles surrounded by electron-dense lysosomes termed lipolysomes [13]. Although these lesions seen in copper overload appear to vary from species to species, the sequestration of copper within lysosomes protecting hepatocytes from its toxicity seems to be a general feature [14]. When we comparatively analyzed liver section from wild type and *Atp7b*^–/–^ mice, we found the typical condensed mitochondria with dense deposits in the *Atp7b*^–/–^ mice (not shown) and many other ultrastructural changes typical for WD reported before [15]. However, most strikingly was the high content of lipolysomes enriched in fat droplets and in electron dense particles with an irregular shape in *Atp7b*^–/–^ liver samples (Figure 2). These conspicuous sphere-shaped vesicles were very irregularly distributed and not found in wild type controls.

### 3.2. Rhodanine Stain for Detection of Cytoplasmic Accumulation of Hepatic Copper in Atp7b^–/–^ Mice

In daily routine work, tissue copper is often assessed semi-quantitatively by rhodanine stain using sections cut from formalin-fixed, paraffin-embedded tissue blocks. To demonstrate the success of copper coloring, a control tissue that is most often taken from a confirmed WD patient is stained in parallel. Usually, the rhodanine method gives the most reliable results with a good correlation between the results of histochemical staining and tissue copper levels [2,16]. However, it is also known that the positive staining for copper and copper-binding proteins is majorly found within lysosomes and can vary considerable throughout the liver. The result of the staining is further influenced by time of fixation, staining, incubation time, and other variations in the procedure. Therefore, it is assumed that this method is applicable as a screening method for the semi-quantitative evaluation of tissue copper [16]. In agreement with this assumption, we successfully used this stain to document the cytoplasmic accumulation of copper in human WD patients (Figure 3A). However, the stain failed when we analyzed liver samples from *Atp7b*^–/–^ mice taken at age 58–102 weeks known to have high quantities of hepatic copper within liver [7,9,17], while in contrast, the positive control taken from a human WD patient showed the typical lysosomal copper deposits (Figure 3B).

### 3.3. Analysis of Liver Tissue of Atp7b^–/–^ Mice by Scanning Electron Microscopy with Energy-Disperse X-Ray Spectroscopy Analysis

Scanning electron microscopy (SEM) with energy-disperse X-ray (EDX) spectroscopy analysis (SEM-EDX) is an elemental microanalysis technique commonly used for chemical analysis and determination of the elemental composition within a specimen or a tissue. It is capable to detect metal deposits by in situ measuring the energy and intensity distribution of X-ray signals generated by a focused electron beam microscopic in a liver biopsy. After a challenge with X-rays, element-specific peaks on the X-ray spectrum appear allowing discriminating of individual metals including cadmium (Cd), chromium (Cr), copper (Cu), lead (Pb), nickel (Ni), zinc (Zn) and many others [5]. This technology is sometimes used in WD diagnostics to detect copper-specific signals of electron-dense accumulations inside lysosomes and residual bodies [18].

However, when we applied this method to *Atp7b*^–/–^ liver samples, it failed to identify copper deposits, while it allowed to identify different other elements (Ca, Na, Cl, K, Si, C, O) that were included in the analyzed aggregates (Figure 4).

### 3.4. Detection of Hepatic Copper Overload in Atp7b^–/–^ Mice by Laser Ablation Inductively-Coupled Mass Spectrometry

Recently, we established novel LA-ICP-MS-based protocols for metal bio-imaging in different organs including liver, kidney, and brain [19,20]. We have already comprehensively demonstrated that respective protocols are highly suitable to identify and quantify copper deposits in experimental and clinical samples of WD [6,7,21,22,23]. This technique provides quantitative information about major, minor, and trace elements with wide element coverage and excellent limits of detection [24]. It can determine the concentrations of elements in a linear dynamic range of up to 10 orders of magnitude. Most importantly, LA-ICP-MS is a method for simultaneous multi-element analysis, providing profound information about the spatial resolution of nearly all kinds of elements within the analyzed sample (e.g., liver biopsy). Processing of measured LA-ICP-MS raw data with specialized programs further allows the visualization of regional metal concentrations and differences in colored and vivid imaging maps, which are helpful in the interpretation of data [12].

LA-ICP-MS imaging with subsequent visualization of raw data in ELAI [11] nicely showed increased hepatic copper content in all *Atp7b*^–/–^ animals when compared to wild type controls (Figure 5). While wild type animals showed hepatic copper content of 3.41 ± 0.28 or 3.45 ± 0.30 µg/g liver tissue at 9 or 13 weeks and 2.09 ± 0.28 µg/g liver tissue at 36 weeks, the hepatic copper content was about 112.72 ± 13.26 µg/g liver tissue (Appendix A). Copper was relatively homogenous distributed in wild type and *Atp7b*^–/–^ mice liver and elevated copper content in *Atp7b*^–/–^ mice correlated with a drop in hepatic magnesium and a slight increase in zinc. The iron pattern had a distinct distribution that reflected the typical histology of the liver with its hexagonal organization. This element showed highest concentrations around the rim of the lobule structure. Similarly, manganese showed a somewhat irregular distribution within the liver tissue zones.

## 4. Discussion

The diagnosis of WD is extremely difficult for pathologists and clinicians and is frequently underdiagnosed. Clinical symptoms, such as the formation of Kayser-Fleischer rings and neuropsychiatric disturbances, can have a wide variety [25]. Likewise, clinical measurements such as urinary copper excretion, ceruloplasmin, or total serum copper can be in the normal range. In contrast, hepatic copper accumulation is the hallmark of WD [25]. However, in late stages of Wilson disease, copper is distributed unevenly in the liver, and in some cirrhotic patients, copper can be elevated primarily in only some lobes, suggesting that the determination in a small liver sample cannot be considered as absolutely representative of the mean hepatic copper concentrations.

Commonly used copper stains including rhodanine or orcein detect only lysosomal copper, while copper present in the cytoplasm bound to metallothionein is not histochemically detectable. Therefore, it is estimated that copper histochemistry is only suitable to detect focal copper stores in less than 10% of patients [25]. It is also known that the amount of copper varies from nodule to nodule in cirrhotic liver and may vary from cell to cell in pre-cirrhotic patients [26]. Consequently, the overload with copper cannot be excluded from histochemical evaluation alone because this test has a poor predictive value for WD [26]. Moreover, parenchymal copper is inhomogeneous distributed within the liver in later disease stages and measurement of hepatic tissue copper concentration, which represents a diagnostic gold standard, might be underestimated due to sampling errors [25]. Although the limited use of direct staining of copper in WD was repeatedly illustrated over the last decades [2,4,27], this methodology is still one of the routine methods in WD diagnostics.

Similarly, ultrastructural analysis of liver specimens is suitable for detecting specific mitochondrial abnormalities in size and shape, lipids, and fine granular material, which may be copper or copper-protein aggregates [26,28]. However, electron microscopic analysis only provides detailed black and white images of cellular structures, is prone to errors, expensive, and requires specialist operators, which must undergo years of training to acquire required capabilities. Even when all these hurdles are removed, it is still not possible to obtain element concentrations for a specific element.

SEM-EDX microanalysis enables a local or comprehensive analysis of the most diverse inorganic materials and metal alloys [5]. With this technique, only the relative atomic composition can be determined, but element quantification is difficult [29]. Furthermore, this technique is highly sensible to contamination and the analytical error is higher when structures without constant flat surfaces (e.g., metals deposits) are analyzed. In our study, we failed to demonstrate hepatic copper overload in *Atp7b*^–/–^ mouse by SEM-EDX. In a previous study, this technology was successful in the detection of copper-specific signals of electron-dense accumulations inside lysosomes and residual bodies in two patients suffering from WD, suggesting that this methodology is helpful in WD diagnosis [18]. However, in this report, EDX analysis was only used qualitatively, and final concentrations were measured by atomic absorption spectroscopy. This methodology also has some general limitations. Firstly, the analyst who performs the analysis has a bias and will restrict their analysis to material deposits that are visually detectable. Secondly, several elements have overlapping EDX peaks, preventing the precise assignment of signals to element in some cases. Thirdly, element quantification is often hard to achieve or even impossible. Finally, the accuracy of the final EDX spectrum can be affected by the nature of the sample per se, i.e., the matrix, in the way that inhomogeneous samples result in inadequate or dissimilar excitations.

In our study, we used the *Atp7b*^–/–^ mouse model and showed that LA-ICP-MS overcomes all these problems and limitations in the determination of copper concentrations. When compared to other methods, this technique has many advantages. Most importantly, it provides information about absolute copper concentrations and its distribution within the tissue, irrespective of its oxidation status or its incorporation into protein complexes. This allows scoring copper overload with high diagnostic accuracy and sensitivity, which are both necessary to prevent failures in the diagnosis of WD. The appealing high resolution element distribution maps that can be generated of raw LA-ICP-MS data are helpful in identifying minimal histologic abnormalities characterized by altered element composition. As a multi-element methodology, it can analyze a full range of major, minor, and trace elements simultaneously in one run and tissue sample, including elements such as zinc or iron. In regard to WD, this possibility is relevant because zinc formulations are approved in the US and in Europe for treatment of WD [30]. Although many adverse effects of zinc therapy have been described, zinc formulations are regarded to be an overall safe and well-tolerated drug; thus, the simultaneous measurement of zinc will be helpful for monitoring unwanted side effects induced by zinc overdosage [31]. Similarly, concurrent measurement of iron in the liver specimen will allow identifying early iron overload occurring in patients subjected to standard de-coppering D-penicillamine therapy [32], potentially resulting from a blockade of the iron efflux from the liver to the circulation resulting from hypoceruloplasminemia and resulting reduction of ferroxidase [33,34]. Further advantages of LA-ICP-MS are minimal sample preparation and high sample throughput [24]. These features are prerequisites for routine clinical use.

Recently, synchrotron X-ray fluorescence (XRF), which is also capable to quantify multiple elements, was proposed as another versatile and powerful method for the diagnosis of WD [35]. In the respective study, the authors demonstrated that XRF experiments can be performed on unfixed or fixed tissue sections. However, the diagnosis is based on the estimation of copper content related to iron and zinc (i.e., Cu/Cu + Fe + Zn), which allows discrimination of WD from other liver diseases [35]. Like in our LA-ICP-MS study, XRF allowed to diagnose WD in patients with negative rhodanine staining on liver tissue sections. However, this methodology does not provide quantitative data of element concentrations and or severity of copper overload.

In experimental research, LA-ICP-MS is also useful to assess precise element concentrations in other tissues including the brain. However, it is critical to notice that LA-ICP-MS imaging is done on tissue sections. Therefore, this methodology will not be useful to measure cerebral element concentrations in living humans. In these cases, it will be necessary to apply other imaging techniques such as magnetic resonance imaging (MRI). In some cases, these techniques are also suitable to document the possible connection of copper overload with changes in other elements. This was exemplarily documented in a study investigating a 27-year-old male WD patient, in which the elevated copper concentrations were associated with cerebral iron deposits in treatment-naïve Wilson disease patients, most likely caused by sequestration from dying cells, influx of iron-laden macrophages, increased uptake due to cellular energy production failure, or impaired tissue iron efflux associated with ceruloplasmin dysfunction [36,37].

Compared to the standard methodologies used in the diagnosis of WD such as atomic absorption spectrometry, LA-ICP-MS provides information about element concentrations with their spatial distribution. It will now be essential to convince the clinicians that have used traditional methods for many years that LA-ICP-MS is a significant add-on in WD diagnosis, especially because some traditional methodologies, including rhodanine stain, electron microscopy, and EDX analysis, have only limited diagnostic value.

We must admit more comparative studies are now necessary to finally define if the LA-ICP-MS methodology and workflow are suitable to be incorporated into the routine diagnostics of WD. It will also be interesting to follow if LA-ICP-MS can compete with other sophisticated techniques such as Time-of-Flight Secondary Ion Mass Spectrometry (ToF-SIMS), ambient mass spectrometry imaging with Picosecond Infrared Laser Ablation Electrospray Ionization (PIR-LAESI), and many others. Some elegant studies have already shown that these methods are useful in providing information about distributions of elements/molecules in biological samples with excellent spatial resolution [38,39]. However, none of these methods were actually introduced in the field of diagnostic metal measurements in WD.

In sum, we have shown that LA-ICP-MS is ideally suited to identify and quantify copper overload in WD liver samples using standards for the determination of element concentrations produced from homogenized tissue spiked with varying concentrations of a standard salt solution. Presently, the usage of such in-house prepared “matrix-matched” standards is the dominating strategy in many LA-ICP-MS applications [24]. Establishing more defined reference materials and introducing internal standards will be necessary to avoid pitfalls in normalization and to compare results obtained within different laboratories. Once these problems are solved, LA-ICP-MS will be the most accurate method that will open new avenues in diagnosis and disease monitoring of WD.

## 5. Conclusions

LA-ICP-MS imaging of liver samples is excellently suited to identify and quantify hepatic copper overload in WD. Compared to many other routine tests done for Wilson disease, this procedure provides accurate data on copper and other element concentrations with high sensitivity, spatial resolution, and quantification ability. Based on these features, in the future, LA-ICP-MS imaging will be a versatile methodology in experimental and clinical studies investigating all aspects of metal disturbances in WD pathology. It definitely offers new opportunities for a deeper understanding of the biology behind WD pathogenesis and its copper-associated liver damage. However, limitations for the acceptance of LA-ICP-MS as an alternative to traditional procedures in WD diagnostics, such as the availability of appropriate reference and standard materials that presently prohibit clinical application, will hopefully be solved quite soon.

## Figures and Tables

**Figure 1 biomedicines-08-00356-f001:**
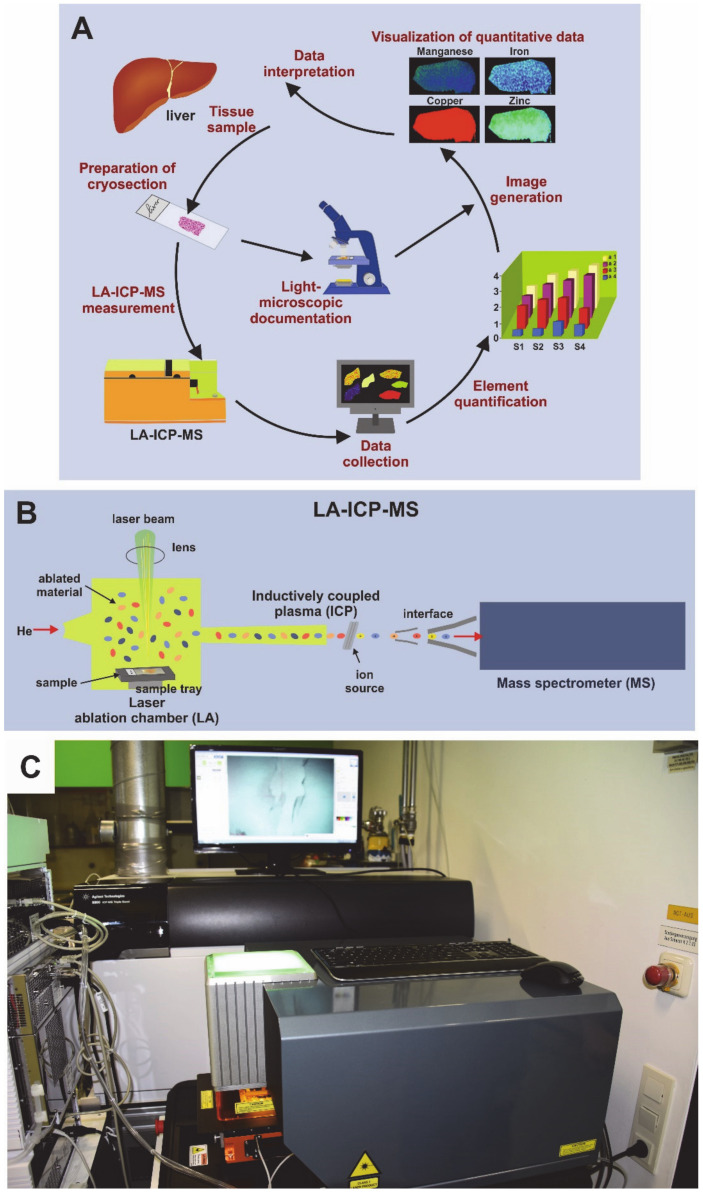
Laser ablation inductively-coupled mass spectrometry (LA-ICP-MS). (**A**) The workflow in LA-ICP-MS consists of different steps including the preparation and documentation of cryosection, the LA-ICP-MS measurement, data collection, element quantification, and the visualization of measured data in element maps that are useful for data interpretation. (**B**) The LA-ICP-MS device is composed of a laser chamber, an inductively coupled plasma (ICP) source, and a mass spectrometer. The individual elements (i.e., metals) are identified by their specific mass to charge (m/z) ratios. (**C**) Shown is a typical setup of an LA-ICP-MS device that was used in this study.

**Figure 2 biomedicines-08-00356-f002:**
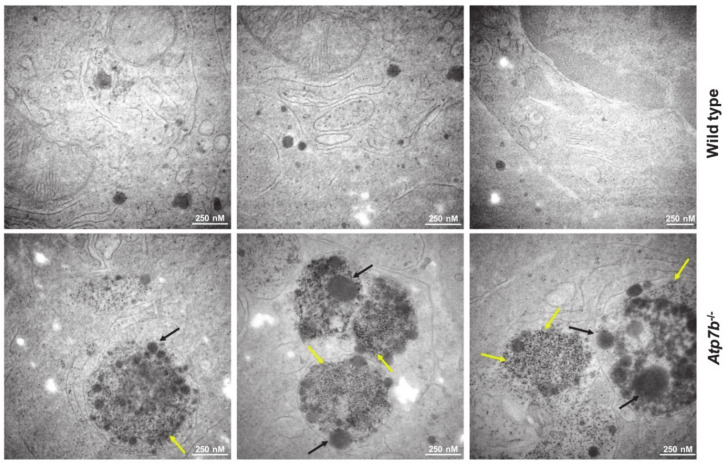
Electron microscopic analysis of liver sections. Representative liver pieces from wild type and *Atp7b*^–/–^ mice were fixed and examined by electron microscopy. The *Atp7b*^–/–^ samples show frequent electron-dense deposits (yellow arrows) and huge fat droplets (black arrows).

**Figure 3 biomedicines-08-00356-f003:**
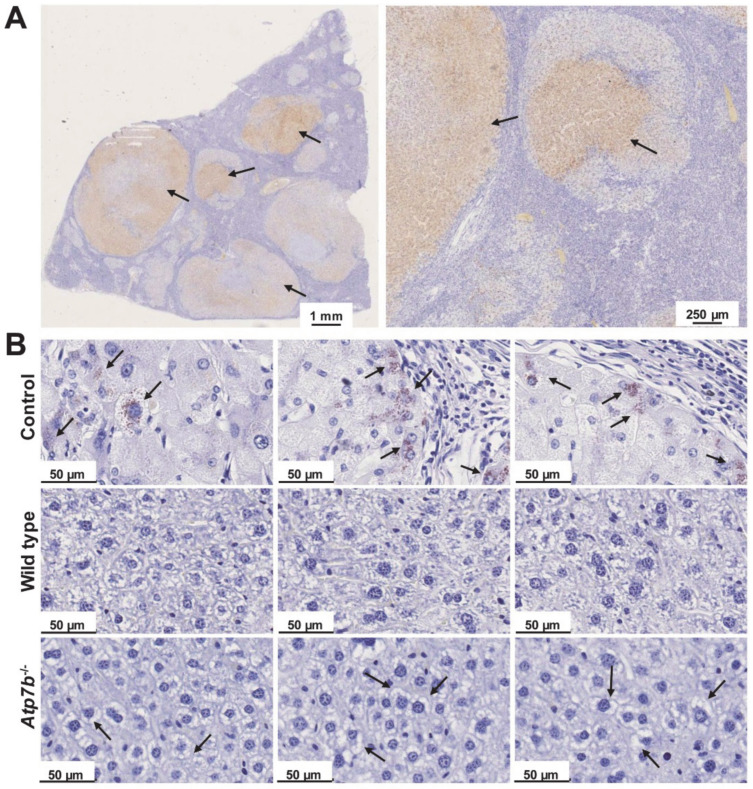
Rhodanine copper stain. (**A**) Rhodanine is a chelating agent forming red-brownish complexes with proteinaceous copper deposits (marked by arrows). Shown is a representative stain of a liver section of a Wilson disease patient at different magnifications. (**B**) However, the classical rhodanine stain yielded false negative results in *Atp7b*^–/–^ mice, showing that the diagnostic value of this histochemical staining is limited. In microphotographs, only the microvescular steatosis in *Atp7b*^–/–^ liver sections (marked by arrows) was noticed.

**Figure 4 biomedicines-08-00356-f004:**
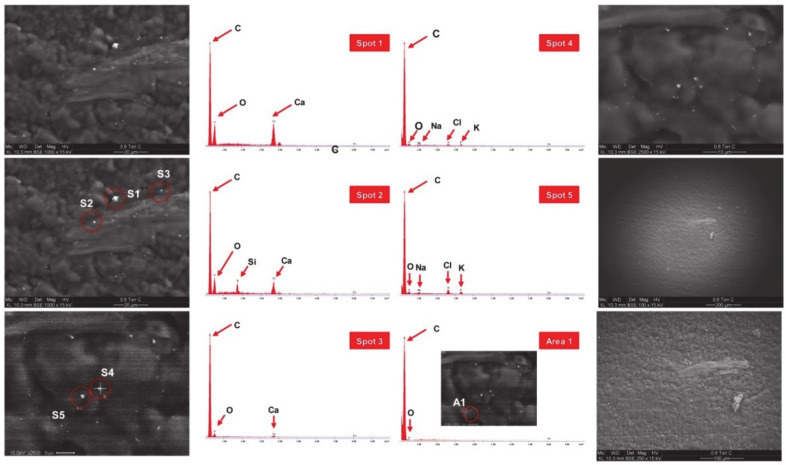
Representative scanning electron microscopy (SEM)/energy–disperse X-ray (EDX) microanalysis of *Atp7b*^–/–^ liver sections. SEM overviews are shown and selected spots (S1–S5) or areas (A1) of minerals analyzed by EDX are marked by red circles. Identified elements are depicted for each spot or area analyzed. In two different set of experiments, copper deposits were not detected in *Atp7b*^–/–^ liver samples.

**Figure 5 biomedicines-08-00356-f005:**
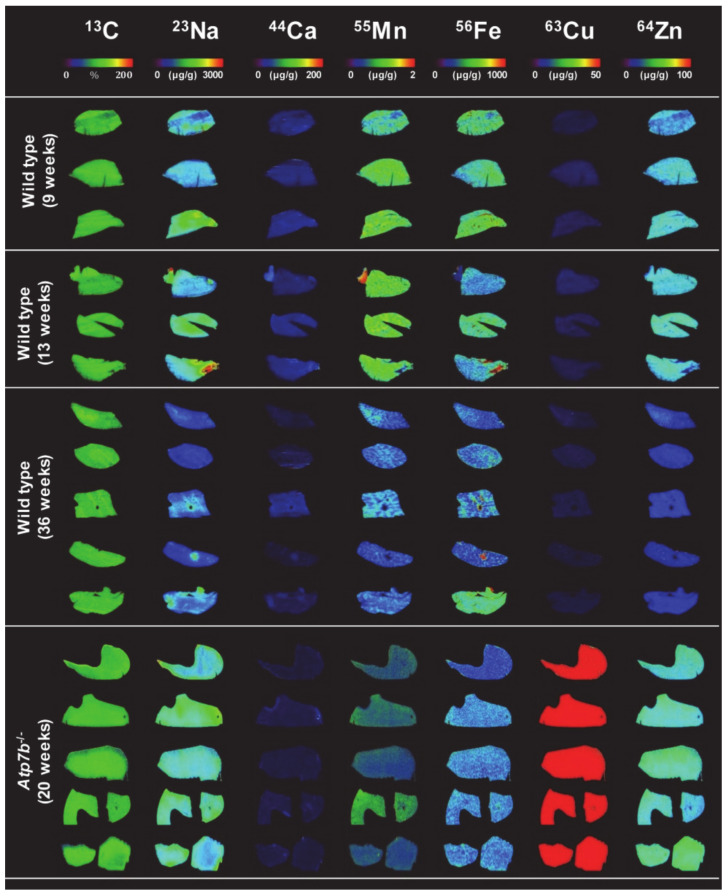
Analysis of the *Atp7b*^–/–^ copper overload model by LA-ICP-MS. Shown are representative LA-ICP-MS images of carbon (^13^C), sodium (^23^Na), calcium (^44^Ca), manganese (^55^Mn), iron (^56^Fe), copper (^63^Cu), and zinc (^64^Zn) in sections from livers of wild type and *Atp7b*^–/–^ mice with indicated age. In the *Atp7b*^–/–^ mice, copper overload is vividly detected by LA-ICP-MS. In this analysis, ^13^C was used for standardization.

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
