# Peer review of "Accurate Measurement of Copper Overload in an Experimental Model of Wilson Disease by Laser Ablation Inductively Coupled Plasma Mass Spectrometry"

_biomedicines, 2020, doi:10.3390/biomedicines8090356_

Round 1

Reviewer 1 Report

It is very well written paper according to metals assesment (especially copper) in  WD tissues with LA-ICP-MS. This issue is very important in WD diagnosis (copper content in liver) as well as research according to significant elf other metals involvement in WD progression (especially iron?)

I know that the article is mainly about liver metal analysis, but according to clinical symptoms of WD, would authors add (just mention) in discussion sections some information if this method would be also suitable to asses the other tissues including brain (research in WD). Currently the brain iron accumulation was also found in WD and was proposed as factor which could impact WD presentation (Dusek, et al Neuropathology and Applied Neurobiology 2018)

I recommend to accept this article with this one suggestion

Reviewer 2 Report

The manuscript entitled "Accurate measurment of copper overload in an experimental model of Wilson disease by laser ablation inductively coupled plasma mass spectrometry" is interesting but unfortunately does not bring  a strong novelty to the field.

An other paper published on MDPI Inorganics 2019, 7, 54 by the same authors show  similar results using, among others ,experimental model liver and  real human liver samples. 

In the current study the added a comparison of LA-ICP-MS results with other techniques.  They show results from SEM-EDX array that does not detect copper. It is well-known by the scientific community that EDX is not sensitive enough to allow comparisons with LA-ICP-MS (which is much more sensitive).  The authors also used also rodhanine copper staining that is the gold standard techique showing that the diagnostic values of this technique is limited.

When running comparative studies  with the aim of highlithing the advantages of a new analytical method well-established techniques capable to detect the analyte should be chosen (see as an example the introduction of Picosecond Infrared Laser Ablation Electrospray Ionization mass spectrometry Zou et al Anal. Chem. 2015, 87, 12071−12079 that compares this new technique with the well-established MALDI-MS, DESI-MS and LA-ESI-MS).

I would use TOF-SIMS as comparative method. TOF-SIMS is a well-established method for the analysis of metals in different types of surfaces and tissues. See as an example the following paper 10.1016/j.apsusc.2006.02.227

I would also suggest to provide data about the robustness of the method such as limit of detection, linearity of the response and accuracy.

The photo in Figure 1 does not provide any information about the technique. If the authors want to show how the technique works,  they could provide a scheme of the ionization method or a scheme of the workflow. 

The English is poor. It must be revised.

Round 2

Reviewer 2 Report

I am pleased with the changes.